# Learning without training:
# The implicit dynamics of in-context learnings

## Abstract

One of the most striking features of Large Language Models (LLMs) is their ability to learn in-context. Namely at inference time an LLM is able to learn new patterns without any additional weight update when these patterns are presented in the form of examples in the prompt, even if these patterns were not seen during training. The mechanisms through which this can happen are still largely unknown. In this work, we show that the stacking of a self-attention layer with an MLP allows the transformer block to implicitly modify the weights of the MLP layer according to the context. We argue through theoretical analysis and experimentation that this simple mechanism may help explain why LLMs demonstrate capabilities of in-context learning, beyond what is captured during training. Specifically, we show that a standard forward pass with context is mathematically equivalent to a forward pass without context but with the MLP weights updated by a minimal low-rank update representing the context.

## 1 Introduction

Large language models (LLMs), powered by the transformer architecture (Vaswani et al., 2017), have revolutionized modern machine learning, with wide-ranging applications in science, industry, and art (Liu et al., 2023; Dong et al., 2024). Despite this impact, the mechanisms behind their impressive emergent properties (Wei et al., 2022; Bubeck et al., 2023) are still not fully understood. One of the most fascinating and compelling of these properties is the ability of LLMs to perform in-context learning (ICL) wherein the model is able to adapt based on information provided in the input prompt, without any changes or modifications to the model's underlying weights. Our work is focused on better understanding the mechanisms which enable this advantageous behavior.

Historically, in machine learning, the ability to extract patterns from data has been understood as a dynamical process in which model weights are updated through an optimization procedure (Goodfellow et al., 2016). However, in the case of ICL, the model weights remain unchanged. Instead, LLMs appear to re-organize or reconfigure their internal representations depending on the prompt and this dynamic adjustment allows them to make predictions that are significantly more accurate. This mysterious and extremely helpful property of LLMs has led researchers to conjecture an implicit form of weight updates taking place at inference time when a prompt is consumed (Garg et al., 2022; von Oswald et al., 2023; Dai et al., 2023; Akyürek et al., 2023; Zhang et al., 2024; Huang et al., 2025). Recent works have even been able to theoretically justify this intuition, showing that simplified transformer blocks, trained on toy setups of linear regression datasets, perform implicit weight updates corresponding to a sort of gradient descent optimization (von Oswald et al., 2023; Dai et al., 2023; Zhang et al., 2024). Together, these works suggest it is possible to understand ICL as a form of implicit finetuning of the original pretrained model. In this work, we follow this intuition of ICL as imposing implicit weight updates and focus on the contextual information property which we believe is key to understanding the underlying effect of ICL. To this end, we introduce the notion of a contextual block, a generalization of a transformer block. We show that layers with this contextual property, when stacked with standard neural networks, implicitly transform a context into a weight update of the very first layer of the subsequent neural network. Through our analysis we are able to provide an explicit formula for this implicit update to the feedforward layer weights, which surprisingly turns out to be a rank-1 matrix. Interestingly,

other works such as Meng et al. (2022) have uncovered that explicit updates with similar rank-1 matrices can modify factual information in a LLM. This suggests that these low-rank matrices may be central to the way LLMs organize and process information at inference time.

Namely, our work demonstrates that contextual blocks, such as self-attention layers combined with a neural network, perform a sort of *implicit* low-rank finetuning that can be explicitly described as a minimal token-patch: a unique, Frobenius-norm minimizing, rank-1 matrix update of the MLP weights computed directly from the relative effect on the context.

Let us clarify what we mean by *implicit.* Our perspective differs from mechanistic approaches that analyze the explicit computation of a forward pass directly, for instance in terms of circuits (Elhage et al., 2021; Olah et al., 2020; Conmy et al., 2023). Instead, we show that a forward pass with a context is *functionally equivalent*—identical in output—to a *different computation* that is never performed on the hardware: one in which a weight update replaces the prompt context. Because this weight update is never directly computed but only equivalent to the standard forward pass, we call these contextual weight updates *implicit.* Likewise, the learning dynamics we uncover is not executed as a computation on the hardware; the actual output is merely mathematically equivalent to what would result if such a learning process had taken place. This is an abstract viewpoint of mathematical equivalence, analogous to studying discrete optimization through continuous dynamical systems.

Our main contributions are as follows:

- We introduce the notion of a contextual block which generalizes the information-processing mechanisms found in architectures such as transformers.

- We define the minimal token-patch and show that for contextual blocks the context acts as an implicit rank-1 update of the MLP weights, uniquely characterized as the Frobenius-norm minimizing solution.

- Using these implicit updates, we uncover an implicit gradient descent learning dynamics which arises as prompt tokens are consumed during inference.

Note that our implicit weight update formula has two parts in the context of transformer blocks with skip connections (see Theorem C.2): a low-rank weight matrix update and a vector update. The former is reminiscent of the updates found in factual knowledge editing like in Mitchell et al. (2022) or Meng et al. (2022), while the latter has strong similarities to the steering vectors found in Ilharco et al. (2022), Hendel et al. (2023), or Todd et al. (2023) for instance. In a way, our work connects steering vectors and low-rank matrix edits to the implicit mechanisms of the transformer architecture.

## 2 Implicit update theory

In this section, we abstract some key properties of transformers. In particular, we introduce the notion of a contextual layer, which generalizes the self-attention layer found in transformer blocks. In this setting a contextual block is the composition of a contextual layer with a standard neural network generalizing the notion of a transformer block. Then we prove our main theorem, which shows that the context for contextual blocks acts as a low-rank fine tuning update of the neural network weights. Based on this, we derive the implicit learning dynamics of ICL. We also discuss a natural application to prompt compression in Section 3.4. For the sake of simplicity, we state our results in the case of a neural network without skip-connection, and for one contextual block. The skip-connection case with multiple contextual blocks is similar but more complicated and fully worked out in Appendix C.

### 2.1 Contextual Blocks and Minimal Token Patches

We define a *contextual layer* as a network layer $A(\cdot)$ that can take a single vector $x$ alone as input yielding an output $A(x)$; or, optionally, $A$ can take in addition a context $C$ (e.g., a sequence of tokens, an image, etc.) along with the vector $x$, yielding the output.

As a prototypical and guiding example of a contextual layer, consider the self-attention layer of a transformer block, where the context $C$ is an instruction prompt consisting of a sequence of context tokens $C = [c_1, \ldots, c_n]$ and $x$ is the query token from which the LLM will make a prediction. Together $C$ and $x$ create a contextualized input prompt $[C, x] = [c_1, \cdots, c_n, x]$, which is the concatenation of the context tokens and the query token. Note that a transformer maps sequences of a given length to a sequence of the same length. Therefore, we take $A(C, x)$ to be the output of the self-attention layer corresponding to last token $x$[1]. In this way, both $A(C, x)$ and $A(x)$ occupy the same output vector space.

**Definition 2.1.** A *contextual block* is the composition $T_W = M_W \circ A$ consisting of a contextual layer $A$ as above with a neural network $M_W$; i.e., $M_W(z) = f_\theta(Wz + b)$, where $W$ and $b$ are the weights of an initial fully-connected dense layer and $f_\theta(z)$ is the rest of the neural network parameterized by weights $\theta$.

In what follows, we demonstrate that the contextual effect of any sequence $Y$ can be perfectly assimilated into the static weights $W$. Let $Y$ be a contiguous subsequence of the context $C$ (e.g., a set of in-context demonstrations). We denote the ablated context, where $Y$ has been removed, as $C \setminus Y$. Formally, given a context $C$ and a query token $x \in C \setminus Y$, we ask whether there exists a weight displacement $M$ such that the output of a forward pass over the partial context $C \setminus Y$ using modified weights $(W + M)$ is functionally identical to the standard forward pass over the full context $C$.

Because the non-linear network $f_\theta$ is not generally injective, finding all matrices $M$ that satisfy $T_{W+M}(C \setminus Y, x) = T_W(C, x)$ is intractable and ill-posed. Instead, we seek a principled solution by demanding *pre-activation equivalence*. We seek matrices $M$ that exactly match the linear input representation to $f_\theta$: i.e., $(W + M)A(C\setminus Y, x) = WA(C, x)$. This requires satisfying the *representation-matching equation*

$$MA(C \setminus Y, x) = W\delta A_x(Y), \quad \text{where} \quad \delta A_x(Y) := A(C, x) - A(C \setminus Y, x) \tag{1}$$

is the *context vector* for chunk $Y$. The representation-matching equation trivially guarantees functional equivalence: Namely

$$
\begin{aligned}
(W + M)A(C \setminus Y, x) &= WA(C \setminus Y, x) + MA(C \setminus Y, x) \\
&= W(A(C \setminus Y, x) + \delta A_x(Y)) \\
&= WA(C, x).
\end{aligned}
$$

Among all solutions $M$, we seek the one that modifies the existing weights minimally:

**Definition 2.2** (Minimal Token-Patch)**.** For a context chunk $Y$ and a query token $x$, a *token-patch* is any matrix $M_x \in \mathbb{R}^{d \times d}$ that guarantees functional equivalence by solving the representation-matching equation $M_x A(C \setminus Y, x) = W\delta A_x(Y)$. We define the *minimal token-patch* as the unique solution in this class that strictly minimizes the Frobenius norm $\|M_x\|_F$.

The following theorem provides a general formula for the unique minimal token-patch.

**Theorem 2.3.** *Consider a contextual block $T_W = M_W \circ A$ formed by a contextual layer $A$ composed with a neural network $M_W$ whose first fully-connected layer has weight matrix $W$. Given a context $C$ and an input $x \in C \setminus Y$, provided that $A(C \setminus Y, x) \neq 0$, there exists a unique minimal token-patch $\Delta_x W(Y)$. In other words, among all matrices that achieve the functional equivalence*

$$T_W(C, x) = T_{W + \Delta_x W(Y)}(C \setminus Y, x) \tag{2}$$

*via pre-activation matching, $\Delta_x W(Y)$ uniquely possesses the minimal Frobenius norm $\|\Delta_x W(Y)\|_F$. It is given by the exact formula:*

$$\Delta_x W(Y) = \frac{(W\delta A_x(Y))A(C \setminus Y, x)^T}{\|A(C \setminus Y, x)\|^2}, \tag{3}$$

*where $\delta A_x(Y) := A(C, x) - A(C \setminus Y, x)$ is the context vector. Furthermore, this unique **minimal displacement update** is conservative: it preserves the original weights $W$ in all directions orthogonal to $A(C \setminus Y, x)$. Specifically, for any vector $u$ such that $u^T A(C \setminus Y, x) = 0$, we have $(W + \Delta_x W(Y))u = Wu$.*

---

[1]The situation our main statements deal with is actually a more general one: Namely, we will want to remove from the context $C$ only a part of it $Y$; in this case the notation $A(C \setminus Y, x)$ or $T(C \setminus Y, x)$ will mean the outputs corresponding to *any* token $x \in C \setminus Y$, and not only the last token in the prompt.

The proof is given in Appendix B.1. The theorem above is a simplified version of our main result which we state here for the sake of simplicity and clarity. We provide the general result and its proof for a stack of contextual blocks *with skip connections* in Appendix C (Theorem C.2).

**Remark 2.4** (Counterfactual interpretation). The context vector $\delta A_x(Y) = A(C, x) - A(C \setminus Y, x)$ is an analytical device measuring the functional effect of the context chunk $Y$ on the layer output. It is not a quantity that the model computes in a single forward pass; rather, it plays a role analogous to counterfactual interventions in causal interpretability (Vig et al., 2020; Conmy et al., 2023), enabling us to isolate and quantify the causal contribution of $Y$ to the model's prediction.

The actual algebraic form of the token patch depends on the architecture of the contextual block. To illustrate this, let us compute the token patch explicitly in the case where $A$ is a linear attention, which gives us some intuition on the nature of these updates:

**Example 2.5** (Linear Attention). When the contextual layer $A$ uses linear attention (i.e., self-attention without softmax normalization), the token-patch can be computed exactly. Let $C$ be a sequence of context tokens with key and value representations $k_i$ and $v_i$, and let $q_x$ be the query representation for token $x$. Let $Y \subset C$ be a context chunk as before. The linear attention output is $A(C, x) = v_x + \sum_i (q_x^T k_i) v_i$, where $v_x$ is the value vector for the query token[2]. In this case, the context vector factorizes as

$$\delta A_x(Y) = \sum_{y \in Y} (k_y^T q_x) v_y = \sum_{y \in Y} (v_y k_y^T) q_x = \Delta(Y) q_x, \quad \text{with } \Delta(Y) := \sum_y v_y k_y^T. \tag{4}$$

Substituting this into our token-patch formula, we see that the token patch also factorizes:

$$\Delta_x W(Y) = W \Delta(Y) P_x \quad \text{where } P_x := \frac{q_x a_x^T}{\|a_x\|^2}, \tag{5}$$

where we used the shorthand notation $a_x = A(C \setminus Y, x)$. Thus for linear architectures, the context chunk $Y$ is completely accumulated into a sort of static $KV$-state matrix $\Delta(Y)$, which is then projected via a rank-1 transition $P_x$ to form the token patch.

## 2.2 The implicit learning dynamics of ICL

With this insight on the relationship between the the context and its implicit affect on the weight parameters, we now use Theorem 2.3 to examine the weight dynamics of $W$ via in-context learning. Namely, when the context $C = [c_1, \ldots, c_n]$ is a sequence of tokens, an iterative application of Theorem 2.3 uncovers an implicit learning dynamics generated by the effect of each context token on the contextual block output. Starting with the initial weight $W_0$ for $M_W$, the first dense layer of the neural network, we compute the weight updates corresponding to the addition of a new context token $c_i$ provided to us by Theorem 2.3. We have iteratively

$$T_{W_0}(c_1, x) = T_{W_0 + \Delta_x W_0(c_1)}(x), \quad T_{W_0}(c_1, c_2, x) = T_{W_0 + \Delta_x W_0(c_1, c_2)}(x), \ldots$$

This leads to the following sequence of corresponding MLP weights

$$W_1 = W_0 + \Delta_x W_0(c_1), \quad W_2 = W_0 + \Delta_x W_0(c_1, c_2), \ldots, \quad W_n = W_0 + \Delta_x W_0(c_1, \ldots, c_n). \tag{6}$$

By construction, this sequence converges to the effect of the full context on the initial MLP weights so that

$$T_{W_0}(c_1, \ldots, c_n, x) = T_{W_n}(x).$$

The following proposition shows that this implicit learning dynamics is similar to that of online gradient descent, where the tokens play the role of the data points and a loss which changes at each step depending of the token considered for that step.

---

[2]Resolving self-attention with a query token $x$ usually yields a term $(q_x^T k_x) v_x$. Here we write $v_x$ for simplicity, implying $q_x^T k_x = 1$, or representing a fixed self-attention weight. In any case, this term cancels out when computing the difference $\delta A_x(Y)$.

**Proposition 2.6.** *In the notation above, the iterative process of weight updates can be realized as a form of stochastic gradient updates $W_{i+1} = W_i - h\nabla_W L_i(W_i)$ with learning rate given by $h = 1/\|A(x)\|^2$ and loss at step i given by $L_i(W) = \mathrm{trace}(\Delta_i^T W)$ where $\Delta_i = W_0\Big(A(c_1, \ldots, c_i, x) - A(c_1, \ldots, c_{i+1}, x)\Big)A(x)^T$.*

The proof is given in Appendix B.1. Notice that $\Delta_i$ measures the marginal effect of the addition of context token $c_{i+1}$ to the partial context $c_1, \ldots, c_i$. Intuitively, when $c_{i+1}$ has no marginal effect on the output, i.e., when $A(c_1, \ldots, c_i, x) = A(c_1, \ldots, c_{i+1}, x)$, we would expect that the corresponding update to the MLP weights $W$ also vanishes. This intuition is quantitatively justified through Proposition 2.6 since by definition $\Delta_i$ indeed vanishes since $A(c_1, \ldots, c_i, x) - A(c_1, \ldots, c_{i+1}, x)$ is zero.

**Remark 2.7** (On the algebraic nature of the gradient descent reformulation)**.** The gradient descent reformulation in Proposition 2.6 is an algebraic consequence of the additive structure of the weight updates: any sequence $W_{i+1} = W_i + G_i$ can be written as $W_{i+1} = W_i - h\nabla_W L_i(W_i)$ for the linear loss $L_i(W) = -\mathrm{trace}(G_i^T W)/h$. The content of the proposition therefore lies less in the gradient descent form per se, but rather in the specific structure of the gradient $\Delta_i$, which measures the marginal effect of incorporating each context token on the attention output. In the linear-attention setting (see below), we show constructively that this formal gradient recovers a task-level least-squares objective for appropriately trained weights. Furthermore, the experimental comparison with finetuning in Section 3 provides empirical evidence that the implicit dynamics tracks a task-relevant objective even for softmax attention. Finally, the alternative dynamics derived in Appendix C offer a complementary sequential perspective on these updates.

**Relation with explicit supervised learning.** The implicit dynamics introduced in Proposition 2.6 is valid for *any* prompt (even those without explicit task data points) and for *any* weight values (including initialization!) of the contextual block. In these settings, there are no tasks, downstream objective, or even meaningful learning dynamics to compare our implicit learning dynamic to. At that level of generality it is perhaps not too surprising that this "meta" dynamics (loss or gradient) is not immediately recognizable as a standard supervised objective. However, when the prompt consists of structured in-context learning demonstrations $C = [c_1, \ldots, c_n]$ in the form of input / output vectors $c_i = (x_i, y_i)^T$ followed by a test query $z = (x, 0)^T$, we have the least-square objective and its gradient descent dynamics to compare with. While establishing an exact analytical equivalence between the implicit dynamics and an explicit least-square dynamics for general non-linear architectures appears analytically intractable (and beyond the scope of this paper), we can do so under simplifying linear attention assumptions standard in the literature (von Oswald et al., 2023; Garg et al., 2022; Akyürek et al., 2023). Namely, in the case of a linear attention layer as in Example 2.5, the implicit gradient in Proposition 2.6 becomes:

$$-h\nabla_W L_i(W_i) = W_0\big(A(C_{i+1}, z) - A(C_i, z)\big)\frac{A(z)^T}{\|A(z)\|^2} = W_0(v_{i+1}k_{i+1}^T q_z)\frac{v_z^T}{\|v_z\|^2} = W_0(v_{i+1}k_{i+1}^T)P_z \qquad (7)$$

where $P_z = \frac{q_z v_z^T}{\|v_z\|^2}$ is the transition matrix. Now, following von Oswald et al. (2023), the question is whether there exist special weight values that the network could learn during training, making the implicit gradient above coincide with a standard least-square gradient. In fact, there are. If we assume for the sake of tractable computation that the MLP consists of only one linear layer $W = [\omega_x, \alpha_y]$ directly predicting the scalar regression target $\hat{y}(x) = [\omega_x, \alpha_y]A(C, z)$, then we can choose $W = [\omega_0, -1]$, where $\omega_0$ is randomly initialized. For the linear attention matrices we choose: $W_V = \mathrm{id}$ (so $v_i = c_i$), $k_i = W_K c_i = (-\eta x_i, 0)^T$ (where $\eta$ is a parameter), and $q_i = W_Q z = (x, 0)^T$. With these weight values, we have that

$$-h\nabla_W L_i(W_i) = -\eta[\omega_0, -1]\begin{bmatrix} x_{i+1} \\ y_{i+1} \end{bmatrix}x_{i+1}^T P_z = -\eta(\omega_0 x_{i+1} - y_{i+1})x_{i+1}^T P_z = -\eta\nabla_{\omega_0}\mathcal{LS}_i(\omega_0)P_z \qquad (8)$$

where $\mathcal{LS}_i(\omega_0) = \frac{1}{2}(\omega_0 x_{i+1} - y_{i+1})^2$ is the standard Least Squares loss for the $(i+1)$-th sample. We can now tie the full dynamics together by considering the weight $W_n$ obtained after processing the entire context $C = [c_1, \ldots, c_n]$. Since each partial update is computed relative to the initial weights $W_0$, the weights accumulate as:

$$W_n = W_0 - \eta\sum_{i=0}^{n-1}\nabla_{\omega_0}\mathcal{LS}_i(\omega_0)P_z = W_0 - \eta\nabla_{\omega_0}\mathcal{LS}(\omega_0)P_z \qquad (9)$$

where $\mathcal{LS}(\omega) = \sum \mathcal{LS}_i(\omega)$ is the full-batch Least Squares loss. We then evaluate the context-free output $T_{W_n}(z) = W_n z$:

$$
\begin{aligned}
T_{W_n}(z) &= (W_0 - \eta \nabla_{\omega_0} \mathcal{LS}(\omega_0) P_z) z \\
&= [\omega_0, -1] \begin{bmatrix} x \\ 0 \end{bmatrix} - \eta \nabla_{\omega_0} \mathcal{LS}(\omega_0) \left( \frac{xz^T}{\|z\|^2} \right) z \\
&= (\omega_0 - \eta \nabla_{\omega_0} \mathcal{LS}(\omega_0)) x
\end{aligned}
\tag{10}
$$

We conclude that in this simplified setting, the implicit learning dynamics is exactly equivalent to one step of full-batch Gradient Descent with learning rate $\eta$ on the weights $\omega_0$. This establishes an existence proof that the model has the capacity to implement gradient descent via its weights, consistent with the observations in von Oswald et al. (2023).

## 3 Experiments

To evaluate our theoretical framework, we conduct a series of experiments in the well-studied setting of in-context learning of linear functions (Garg et al., 2022; Akyürek et al., 2023; von Oswald et al., 2023). All model architecture and training hyperparameters are detailed in Appendix D. Our experiments are designed to achieve four goals: (1) verify that our exact algebraic identities hold under finite-precision arithmetic in deep networks, (2) show these implicit dynamics closely track the explicit dynamics seen during finetuning, (3) demonstrate how our framework serves as an architectural diagnostic tool, and (4) prove practical utility by approximating token-dependent updates into a static "thought patch" for prompt compression.

### 3.1 Numerical Verification of Functional Equivalence

While Theorem C.2 provides an exact algebraic identity for functional equivalence, floating-point arithmetic and ill-conditioned matrices in deep, non-linear networks can cause theoretical identities to break. To assess the robustness of our implicit updates, we conduct our experiments using a 10-layer transformer equipped with LayerNorm and residual connections as in (Vaswani et al., 2017), trained on linear regression.

Using the recursive formulas from Theorem C.2, we compute the implicit weight updates $\Delta W^{(i)}$ and bias updates $\Delta b^{(i)}$ for all layers. Figure 1 confirms that the standard forward pass with the context, $T_{\mathbf{W},\mathbf{b}}(C, x)$, is numerically indistinguishable from the context-free forward pass with patched weights, $T_{\mathbf{W}+\Delta\mathbf{W},\mathbf{b}+\Delta\mathbf{b}}(x)$. The intermediate block outputs agree to a precision of $10^{-6}$, confirming our theoretical framework perfectly captures the mechanics of the forward pass in deep networks.

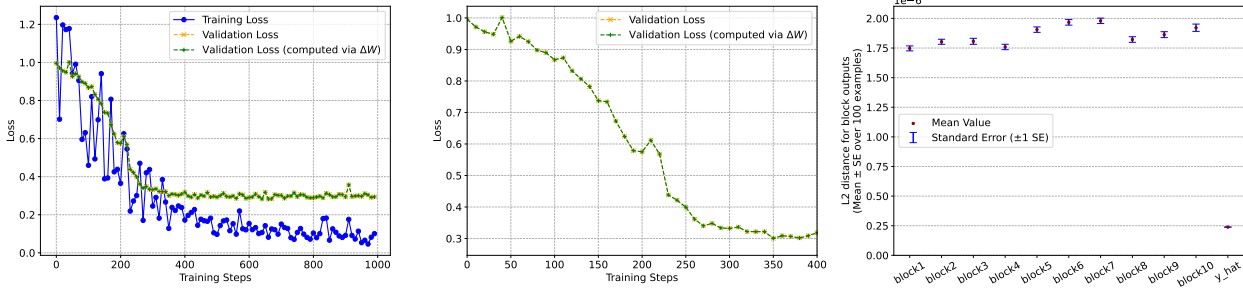

Figure 1: **Numerical Verification in 10-layer Pre-LN Transformer. Left/Middle:** Training and validation loss curves computed via the standard forward pass $(T_W)$ versus the patched context-free pass $(T_{W+\Delta W})$. **Right:** The mean $L_2$-norm difference between intermediate layer outputs across 100 test tasks. Predictions agree to an order of $10^{-6}$.

### 3.2 Implicit Dynamics vs. Explicit Finetuning

To demonstrate that these implicit updates constitute task-relevant learning rather than mere algebraic coincidences, we compare the implicit dynamics of ICL (Prop. 2.6) against explicit Stochastic Gradient Descent (SGD) finetuning.

Using a pretrained transformer, we process a context $C_i = [c_1, \ldots, c_i]$ with $c_l = (x_l, y_l)^T$ of length $i$, computing the exact implicit weight transfer $\Delta_x W(C_i)$ to predict a query token $z = (x_{query}, 0)^T$. In parallel, we initialize the model with pretrained weights and explicitly finetune the MLP weights via SGD on the same $i$ context examples. As shown in Figure 2, both the implicit dynamics and explicit finetuning minimize the test loss in remarkably similar ways. Furthermore, evaluating the normalized Frobenius inner product reveals that the implicit weight updates and explicit SGD gradients remain highly aligned in weight space.

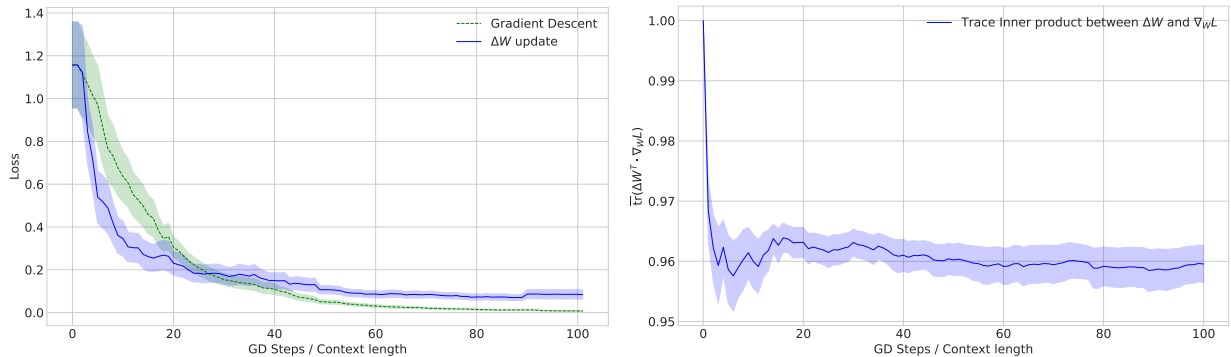

Figure 2: Direct finetuning vs implicit weight update. **Left:** Both finetuning and implicit weight updates minimize the loss in similar ways. **Right:** The two forms of weight updates remain highly aligned with respect to the normalized Frobenius inner product (i.e., the cosine of the angle between the fine-tuned and the implicit gradients.

### 3.3 Architectural Diagnostics: Attention vs. RNN

Because our framework abstracts the core mechanism as a generic "contextual layer" interacting with an affine transformation, it serves as a universal diagnostic tool to evaluate different architectures. By swapping the multi-head attention layer for a Recurrent Neural Network (RNN) layer of comparable parameter count, we can analyze why certain architectures excel at ICL while others struggle. Figure 3 plots the marginal gradient updates $\|\Delta W_{i+1} - \Delta W_i\|_2$ as the context length increases. For Attention-based layers, the implicit gradient updates smoothly decay and converge to zero, indicating stable assimilation of the context. Conversely, the RNN's implicit updates fail to converge, exhibiting highly unstable dynamics. This provides a mechanistic explanation for the empirical observation that standard RNNs struggle with ICL compared to Transformers: they fail to generate convergent implicit learning dynamics in weight space.

### 3.4 Practical Utility: Static Thought Patches via Least-Squares Approximation

Although the goal of this paper is mainly theoretical, establishing the mathematical equivalence between context and implicit weight updates, we consider in this section directions toward practical applications of our theory to prompt compression and inference efficiency. Namely, as noted in Theorem 2.3, the exact minimal token-patch $\Delta_x W(C)$ for a given context $C$ is inherently dependent on the specific query token $x$, making this implicit update not readily useful for prompt compression, as the update needs to be recomputed for every new generated token. To overcome this query specific dependence and achieve practical prompt compression (potentially enabling $O(1)$ inference without a KV-cache), we aim to approximate these instance-specific updates with a single static update $\Delta(C)$ for the context $C$. To achieve this, we define a representative "calibration set" consisting of $K$ queries $\{x_k\}_{k=1}^K$ for a given context $C$. We compute the corresponding exact token-patches $\Delta_{x_k} W(C)$ and pre-activations $a_k = A(x_k)$. We then define the static "Thought Patch,"

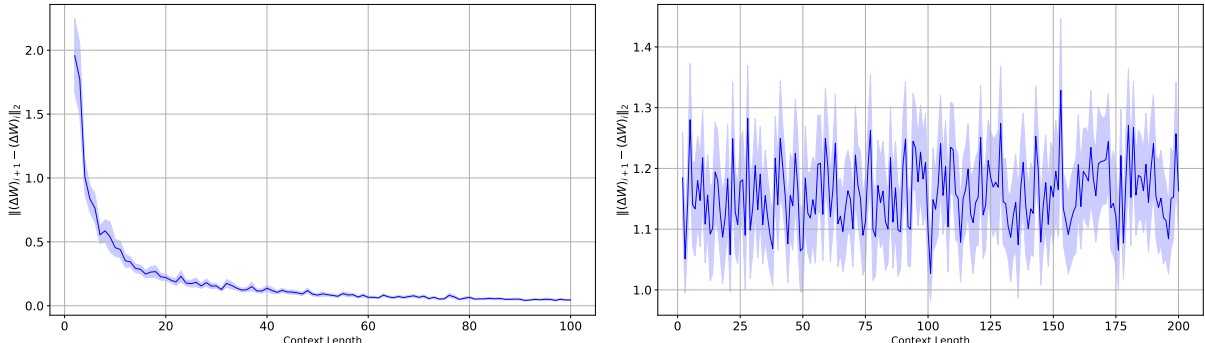

Figure 3: **Architectural Diagnostics. Left:** Attention-based contextual layers produce smooth, converging implicit weight updates. **Right:** RNN-based contextual layers produce highly unstable, non-converging updates, explaining their relative poor ICL performance.

$\Delta(C)$, as the solution to the following optimization problem:

$$\Delta(C) := \arg \min_{\Delta} \sum_{k=1}^{K} \|\Delta \cdot a_k - \Delta_{x_k} W(C) a_k\|_2^2. \tag{11}$$

By solving this objective, we consolidate the diverse token-specific updates into a single weight patch that best approximates the model's behavior across the entire calibration set of size $K$. Once computed, this $\Delta(C)$ serves as a query-agnostic update; meaning it is defined solely by the context $C$ and maintains its utility when applied to query tokens outside the original calibration set.

### 3.4.1 Experimental Setup: Generalization of the Static Thought Patch

To empirically validate the effectiveness of the static "Thought Patch" $\Delta(C)$, we evaluate its performance on a single-layer transformer block trained on linear regression tasks. Specifically, we examine how the accuracy of this query-agnostic approximation converges as the size of the calibration set $K$ increases.

For a fixed context $C$, we define a calibration set $\mathcal{X}_{train}$ of size $K$, where $K \in \{1, 2, 5, 10, 25, 50, 100\}$. We compute the optimal static patch $\Delta(C)$ by solving the optimization problem defined in 11. To assess the generalization capability of $M := \Delta(C)$, we construct a held-out test set $\mathcal{X}_{test}$ of 100 query tokens $x_k$ not contained within $\mathcal{X}_{train}$. To assess the approximation error, we measure the Mean Squared Error (MSE) of the transformer forward pass using the static patch $\Delta(C)$ compared to the ground-truth token-patches $\Delta_x W(C)$. We report the MSE separately for the calibration set $\mathcal{X}_{train}$ (training error) and the held-out test set $\mathcal{X}_{test}$ (test error); see Figure 4 (right). To assess output convergence as $K$ increases, we also measure the fidelity of the patch-based forward pass; see Figure 4 (left). That is, we compute the discrepancy between the full forward pass of the original model output for queries $q$ with the context $C$ and compare with the model output for queries $q$ alone but with with weights modified by $\Delta(C)$; i.e.,

$$\frac{1}{K} \sum_{x_k \in \mathcal{X}_{\{train, test\}}} \left| T_W(C, x_k) - T_{W + \Delta(C)}(x_k) \right|.$$

We note that as $K$ increases, there is a consistent narrowing of the gap between performance on the calibration set and the test set. The convergence of these metrics demonstrates that the static patch $M = \Delta(C)$ transitions from an overfit solution for specific queries (at low $K$) to a robust, query-agnostic approximation that generalizes to unseen tokens (for large values of $K$). This confirms that the "Thought Patch" successfully captures the structural influence of the context $C$ on the transformer's weights.

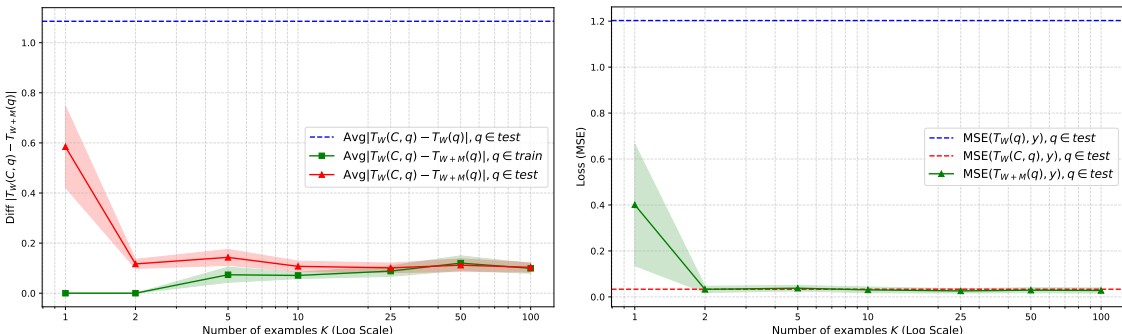

Figure 4: **Generalization of the static "Thought Patch"** $\Delta(C)$ **across calibration set sizes** $K$ . **Left:** Mean absolute difference between the full context-dependent forward pass $T_W(C, q)$ and the patched forward pass $T_{W+M}(q)$, here $M := \Delta(C)$. As $K$ increases, the patched output converges to the full forward pass for both training and held-out test queries. **Right:** The Mean Squared Error (MSE) for the static patch effectively approximates the performance of the full context-aware model, approaching the accuracy of $T_W(C, q)$ (red dashed line) even for unseen test queries as $K$ grows.

## 4 Conclusion and future directions

Our approach to uncovering the transformer's in-context learning mechanics improves upon previous methods in that it does not put any restrictions on the self-attention layer architecture. While earlier theoretical works have also derived a similar form of implicit learning dynamics, these did so only under limiting assumptions on the self-attention layer, such as requiring linear attention or a single head as well as specific forms of prompts; see von Oswald et al. (2023), Dai et al. (2023), and Huang et al. (2025), see also (Shen et al., 2024; Deutch et al., 2024). In contrast our main Theorems (Thm. 2.3 and Thm. C.2) remain valid if the self-attention layer is switched by other forms of contextual layers, such an RNNs, state space models, or any layer that can take an input and optionally a context. This is surprising because our analysis seems to hint that ICL is not so much about the internals of self-attention, but rather about the fact that regular neural networks can transfer modification of input space to their weight structure. Although this is true, and while all contextual layers lead to an implicit weight dynamics as outlined in Prop. 2.6 and Appendix C.2, not all implicit dynamics are equal, and, here, the architecture matters: For instance, our diagnostic experiment shows superior convergence for attention-based contextual layer over RNN-based ones (see Section 3.3.

We conclude by outlining five domains where our theoretical framework may offer significant implications beyond a mechanistic explanation of in-context learning.

First, our results (Theorems 2.3 and C.2) provide a unifying theory connecting ICL to model editing. We demonstrate that *steering vectors* (Ilharco et al., 2022; Hendel et al., 2023; Todd et al., 2023) and *rank-1 factual model edits* (Meng et al., 2022; Mitchell et al., 2022) naturally emerge as distinct aspects of a single phenomenon: the implicit low-rank weight modulation induced by context. This suggests that these heuristic editing techniques are actually intervening on the same fundamental mechanism that the model uses for self-adaptation.

Second, factorization formulas such as Eq. 33 in Appendix C, derived from Theorem 2.3, rigorously map prompt segments to linear operators and textual concatenation to operator composition. We believe that inspecting the algebraic properties of these operators—such as invertibility and commutativity—lays the groundwork for a formal theory of prompt engineering, moving beyond trial-and-error to a principled understanding of prompt interaction.

Third, because our theory enables the extraction of the exact meta-gradient associated with generation (Proposition 2.6), it provides a novel tool for mechanistic interpretability. Monitoring these gradients dynamically could yield valuable insights into generation health, potentially serving as an early detection signal for hallucinations or mode collapse.

Fourth, analyzing ICL dynamics across different architectures suggests a new direction for model design. By evaluating how different "contextual layers" facilitate or hinder implicit weight updates, our framework can guide the development of more efficient architectures optimized specifically for in-context adaptability.

Finally, our results highlight a fundamental distinction between the dynamic nature of attention-based inference and the static nature of standard fine-tuning. As noted in our comparison with Chen et al. (2024), our derived weight updates $\Delta_x W$ depend on the specific query token $x$, suggesting that for general transformer blocks, the context cannot be *exactly* represented by a single, fixed weight update. In Section 3.4, we show how these token-dependent updates can be aggregated—via a least-squares approximation—to produce a single, static weight update (a *thought patch*) that approximates the context for any input. Developing and scaling such approximations bridges the gap between our exact mechanistic description of inference and practical techniques for prompt compression and efficient context reuse.

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

# A   Related work

**In-Context Learning.**   Large language models have the capability to adapt their output based on information or examples provided in the prompt. Because this occurs during inference, there are no explicit gradient updates or modifications of the model parameters. This emergent capability is called in-context learning (ICL) and it has already been shown to exist for GPT-3 in Brown et al. (2020) for a wide range of NLP tasks.

Many works have investigated the behavior of ICL to better understand its underlying mechanisms, often through the lens of meta-learning, or learning-to-learn (Schmidhuber, 1987; Hochreiter et al., 2001; Kirsch and Schmidhuber, 2021). A central question within this research area revolves around the precise nature of the "learning" that takes place during ICL and whether ICL represents genuine few-shot learning or instead serves as a mechanism for task-specific inference steering. For instance, the authors of Reynolds and McDonell (2021) question whether true learning occurs at inference time in ICL, contending that the in-context examples instead help the model retrieve capabilities which were already learned during pretraining. This suggests that no new learning actually takes place at inference time. Specifically, Xie et al. (2022) argues that the examples in the prompt serve only as a form of Bayesian conditioning rather than true learning, and they formalize ICL as Bayesian inference. Supporting this direction, Min et al. (2022) shows that replacing example labels with random labels does not dramatically decrease ICL performance, which bolsters the argument that pretrained capabilities are retrieved from the prompt. Though, revisiting these ideas, Wei et al. (2024) show that these results may in fact vary depending on model size and that larger models do start to actually learn from switched labels within the prompt. Still others (e.g., Raventos et al. (2023)), claim that the emergence of true ICL in large language models seems to be dependent on data diversity during pretraining.

**Gradient Descent and Meta-Optimization.**   A prominent hypothesis is that ICL performs a type of meta-optimization or implicit gradient descent, essentially finetuning the model through the forward pass (Garg et al., 2022; Akyürek et al., 2023; von Oswald et al., 2023; Dai et al., 2023; Ahn et al., 2023; Zhang et al., 2024). Building on this, our work investigates how ICL is implemented through implicit weight updates that correspond to the underlying learning dynamics of the transformer. Many theoretical analyses of these learning dynamics rely on simplifying assumptions such as single-head, linear attention transformers and prompts formatted as input-output examples determined by a fixed function class; e.g., linear regression (von Oswald et al., 2023; Ahn et al., 2023; Zhang et al., 2024). In particular, von Oswald et al. (2023) shows that linear transformers trained on ICL tasks learn to perform updates analogous to gradient descent. Both Zhang et al. (2024) and Ahn et al. (2023) show that transformers can acquire gradient-based algorithms through ICL and prove global convergence to the optimum for tasks like linear regression within this simplified framework. While the formal analysis of Dai et al. (2023) is also derived for linear attention, their empirical results instead focus on large GPT transformers trained on structured language tasks. However, their core conclusion that ICL operates as implicit finetuning resonates strongly with the viewpoint we take on here. Other works investigating the link between ICL and gradient descent for standard transformers have largely focused on prompts structured as input-output pair examples. For instance, Garg et al. (2022) demonstrates that standard transformers can in-context learn diverse function classes from such examples, achieving performance comparable to least squares, while Akyürek et al. (2023) shows they can emulate explicit learning algorithms like gradient descent and ridge regression. This aligns with a broader line of work studying ICL as internal algorithm emulation. For instance, Bai et al. (2023) demonstrates that transformers can act as statisticians by provably selecting and emulating appropriate algorithms in-context, while Hu et al. (2026) explores how models emulate procedures like spectral methods. Furthermore, recent work by Wu et al. (2025) extends the study of in-context deep learning beyond linear models, investigating how transformers implement complex learning processes during inference. Separately, attempts to develop a theory for ICL in standard transformers without restricting prompt structure, such as the work by Liu et al. (2025), have thus far needed to incorporate other architectural simplifications or analytical approximations to make the analysis tractable.

**Task Vectors and Model Editing.** The concept of a task vector in machine learning was first introduced in Ilharco et al. (2022) to describe a direction in a model's weight space that encodes task-specific information. There task vectors are derived from the difference between pretrained and finetuned model weights and the authors show how these vectors could be arithmetically manipulated to effectively steer a model's output. The term has since been expanded to include vectors applied to a model's activations as well and several studies have sought to quantify the effect of ICL by analyzing its influence on both weight task vectors and activation task vectors; notably Mitchell et al. (2022); Meng et al. (2022); Hendel et al. (2023). Similarly, Todd et al. (2023) identify "function vectors" (FVs) in transformer hidden states, which are compact representations of in-context learned tasks, extracted via causal mediation analysis over specific attention head outputs. These FVs are shown to be causally effective in triggering task execution even in novel contexts, and distinct from simple semantic offsets, suggesting they act as higher-level function references within the model. Theoretically our work aligns with and extends these ideas. Namely, we demonstrate that the effect of the context can be precisely mapped to an update of the transformer's parameters. Specifically, we show that this effect can be realized as a direct modification of the feedforward weights. In architectures with residual connections, this also includes an additive bias modification (Theorem C.2) in the final layer, which is functionally equivalent to adding an activation task vector, connecting our weight-centric view with activation-based model editing perspectives.

Among these, the work of Hendel et al. (2023) is particularly relevant and closely related to our own. They show that a transformer maps in-context examples to an "activation task vector" that encodes the underlying rule of the examples provided in the prompt. Similar to our main result, they find that manually adding this task vector to the model's hidden states during inference on a new input (without demonstrations) produces outputs similar to those obtained by manually modifying activations with that task vector. While their work offers a mechanistic view of ICL, our approach differs by theoretically deriving the specific weight and bias modifications equivalent to processing a prompt in-context. We prove that a transformer modified by this weight adjustment yields outputs on new inputs that are identical to the original model's outputs when provided with the in-context demonstrations. Finally, our results offer a theoretical counterpoint to recent efforts in "context compression," specifically the work of Chen et al. (2024). They investigate whether In-Context Learning (ICL) can be explicitly converted into model weights for linear attention transformers. Crucially, they prove that for standard architectures, such exact conversion is mathematically impossible. To circumvent this, they propose a modified architecture that adds special bias terms to the attention layers, allowing the context to be compressed into these new parameters. Our Theorem 2.3 demonstrates that while attention weights may resist such exact compression in standard architectures, the MLP layers do not. We show that the effect of the context can be mapped exactly to a low-rank update of the MLP weights in standard transformers, suggesting that the feedforward network acts as a natural reservoir for context-dependent weight adaptation.

The low-rank nature of our ICL-induced weight update appears in other works which explore techniques for explicit model editing. Most notably, ROME (rank-1 Model Editing) in Meng et al. (2022) injects factual associations into transformers by applying targeted rank-1 updates to feedforward network weight matrices. Similarly, MEND in Mitchell et al. (2022) learns optimized low-rank decompositions for model edits. While these methods engineer or learn low-rank modifications for explicit model editing, our theoretical results show that these rank-1 updates to feedforward weights naturally arise as the mechanism by which transformers implement in-context learning. See also other related works which explore how model editing (either through modification of weights or model activations) can be used to achieve the results of finetuning without any gradient-based learning (Subramani et al., 2022; Panickssery et al., 2023; Li et al., 2023; Zou et al., 2023; Liu et al., 2024; Todd et al., 2024; Uppaal et al., 2024; Yang et al., 2025).

# B  Proofs of the results

## B.1  Minimal update proofs

In this subsection we collect the proofs of the main results from Sections 2.1 and 2.2, as well as supplementary results characterizing the space of equivalent token-patches.

**Proof of Theorem 2.3**

For the reader's convenience, we restate the theorem.

**Theorem B.1** (Restatement of Theorem 2.3). *Consider a contextual block $T_W = M_W \circ A$ formed by a contextual layer $A$ composed with a neural network $M_W$ whose first fully-connected layer has weight matrix $W$. Given a context $C$ and an input $x \in C \setminus Y$, provided that $A(C \setminus Y, x) \neq 0$, there exists a unique minimal token-patch $\Delta_x W(Y)$. In other words, among all matrices that achieve the functional equivalence*

$$T_W(C, x) = T_{W + \Delta_x W(Y)}(C \setminus Y, x)$$

*via pre-activation matching, $\Delta_x W(Y)$ uniquely possesses the minimal Frobenius norm $\|\Delta_x W(Y)\|_F$. It is given by the exact formula:*

$$\Delta_x W(Y) = \frac{(W \delta A_x(Y)) A(C \setminus Y, x)^T}{\|A(C \setminus Y, x)\|^2},$$

*where $\delta A_x(Y) := A(C, x) - A(C \setminus Y, x)$. Furthermore, this unique minimal displacement update is conservative: it preserves the original weights $W$ in all directions orthogonal to $A(C \setminus Y, x)$. Specifically, for any vector $u$ such that $u^T A(C \setminus Y, x) = 0$, we have $(W + \Delta_x W(Y))u = Wu$.*

*Proof.* Let $v = A(C \setminus Y, x)$ and $\delta = W \delta A_x(Y)$. We seek to find the matrix $M$ that minimizes the squared Frobenius norm $\frac{1}{2}\|M\|_F^2$ subject to the linear constraint $Mv = \delta$. Introducing a vector of Lagrange multipliers $\lambda \in \mathbb{R}^d$, we define the Lagrangian:

$$\mathcal{L}(M, \lambda) = \frac{1}{2} \operatorname{Tr}(M^T M) + \lambda^T(\delta - Mv). \tag{12}$$

Taking the gradient with respect to the matrix $M$ and setting it to zero yields:

$$\nabla_M \mathcal{L} = M - \lambda v^T = 0 \implies M = \lambda v^T. \tag{13}$$

Substituting this rank-1 form back into our constraint $Mv = \delta$, we obtain:

$$(\lambda v^T)v = \delta \implies \lambda \|v\|^2 = \delta \implies \lambda = \frac{\delta}{\|v\|^2}. \tag{14}$$

Thus, the unique matrix minimizing the Frobenius norm is $\Delta_x W(Y) = \frac{\delta v^T}{\|v\|^2}$, which matches the formula provided in the theorem. Because the objective function is strictly convex and the constraint is affine, the KKT conditions guarantee this stationary point is the unique global minimum.

To demonstrate the conservative nature of the update, consider any vector $u$ orthogonal to $v$ (i.e., $v^T u = 0$). Applying the patched weights to $u$:

$$(W + \Delta_x W(Y))u = Wu + \frac{\delta(v^T u)}{\|v\|^2} = Wu + 0 = Wu. \tag{15}$$

Therefore, the update leaves the model's behavior strictly unchanged for all inputs orthogonal to the current query representation. $\qquad \square$

**Characterization of equivalent rank-1 updates**

The definition of the contextual block relies on a functional mapping, which introduces a degree of non-uniqueness in the underlying weight parameters. For a given query representation $v_x := A(C \setminus Y, x)$ and a context-induced shift $\delta_x := W \delta A_x(Y)$, any update $\Delta'_x = \Delta_x + M_x$ is functionally equivalent provided $M_x v_x = 0$. This general formulation is unnatural as $M_x$ typically increases the rank of the update and requires an arbitrary choice of a null-space matrix. If we restrict our focus to parsimonious rank-1 updates, the following proposition characterizes the entire equivalence class.

**Proposition B.2** (Characterization of Equivalent Rank-1 Updates). *A rank-1 matrix $\Delta'_x$ satisfies the contextual mapping identity $(W + \Delta'_x)v_x = WA(C, x)$ if and only if it takes the form:*

$$\Delta'_x = \frac{\delta_x w^T}{\langle w, v_x \rangle} \tag{16}$$

*for some vector $w \in \mathbb{R}^d$ such that $\langle w, v_x \rangle \neq 0$. Furthermore, the choice $w = v_x$ yields the unique update with the minimal Frobenius norm $\|\Delta'_x\|_F$.*

*Proof.* By the definition of the contextual block, the update must satisfy $\Delta'_x v_x = \delta_x$. For a rank-1 matrix $\Delta'_x = uv^T$, this implies $u(v^T v_x) = \delta_x$. Thus $u$ must be collinear with $\delta_x$, specifically $u = \delta_x/\alpha$ where $\alpha = v^T v_x \neq 0$. Setting $v = w$ yields the required form.

To minimize the Frobenius norm $\|\Delta'_x\|_F = \|u\|_2 \|w\|_2$, we minimize $\frac{\|\delta_x\|_2 \|w\|_2}{|\langle w, v_x \rangle|}$. By the Cauchy-Schwarz inequality, $|\langle w, v_x \rangle| \leq \|w\|_2 \|v_x\|_2$, with equality if and only if $w$ is collinear with $v_x$. Thus, the norm is minimized when $w = v_x$. $\qquad\square$

## B.2   Implicit dynamics proof

### Proof of Proposition 2.6

For the reader's convenience, we restate the proposition.

**Proposition B.3** (Restatement of Proposition 2.6). *In the notation of Section 2.2, the iterative process of weight updates can be realized as a form of stochastic gradient updates*

$$W_{i+1} = W_i - h\nabla_W L_i(W_i)$$

*with learning rate given by $h = 1/\|A(x)\|^2$ and loss at step $i$ given by*

$$L_i(W) = \text{trace}(\Delta_i^T W)$$

*where*

$$\Delta_i = W_0\Big(A(c_1, \ldots, c_i, x) - A(c_1, \ldots, c_{i+1}, x)\Big)A(x)^T.$$

*Proof.* Considering the sequence of $W_i$'s as defined in Eq. 6 and Eq. 2, the difference $D_i = W_{i+1} - W_i$ between two implicit iterates equals

$$
\begin{aligned}
D_i &= \Delta_x W_0(c_1, \ldots, c_{i+1}) - \Delta_x W_0(c_1, \ldots, c_i) \\[2mm]
&= \frac{W_0\Big(A(c_1, \ldots, c_{i+1}, x) - A(c_1, \ldots, c_i, x)\Big)A(x)^T}{\|A(x)\|^2} \\[2mm]
&= -h\Delta_i,
\end{aligned}
$$

where we set $h := 1/\|A(x)\|^2$ and

$$\Delta_i := W_0\Big(A(c_1, \ldots, c_i, x) - A(c_1, \ldots, c_{i+1}, x)\Big)A(x)^T.$$

This means that

$$W_{i+1} = W_i - h\Delta_i = W_i - h\nabla_W \text{trace}(\Delta_i^T W), \tag{17}$$

since in general we have $\nabla_W \text{trace}(A^T W) = A$. $\qquad\square$

## C Extensions of the Theory

### C.1 Contextual blocks with skip-connections and multiple stacked blocks

We now consider the case of contextual blocks with skip connections, encompassing the standard Pre-LN transformer block as for instance described in He and Hofmann (2024).

**Definition C.1.** A contextual block $T$ with a skip connection is a layer of the form

$$T_{W,b'}(C, x) = \text{LayerNorm}\left(A(C, x) + W' g_\theta(W A(C, x) + b) + b'\right) \tag{18}$$

where $g_\theta$ is any differentiable model parameterized by $\theta$ (or an activation) and $A(C, x)$ is a contextual layer.

Again, our motivation and prototypical example is taken from the standard transformer architecture, where the contextual layer $A(C, x)$ is a multi-head attention block with a skip connection; i.e.,

$$A(C, x) = \text{LayerNorm}(x + \text{MultiHeadAttn}(C, x)).$$

We can generalize Theorem 2.3 to this architecture. Because of the residual connection, functional equivalence requires two modifications: updating the weight matrix $W$ of the first dense layer to match the pre-activations of the MLP, and updating the bias term $b'$ of the final projection to correct the residual stream. Furthermore, we can extend this result rigorously to a sequence of such blocks stacked together.

**Theorem C.2.** *Consider a neural network $T = T^{(L)} \circ \cdots \circ T^{(1)}$ formed by a stack of $L$ contextual blocks $T^{(l)}$ with skip connections as above; i.e.,*

$$T_{W^{(l)}, b'^{(l)}}^{(l)}(C, x) = \text{LayerNorm}\left(A^{(l)}(C, x) + W'^{(l)} g_\theta^{(l)}(W^{(l)} A^{(l)}(C, x) + b^{(l)}) + b'^{(l)}\right) \tag{19}$$

*where $A^{(l)}$ is a contextual layer and $g_\theta^{(l)}$ is a differentiable model. Given a context $C$, an input $x \in C \setminus Y$, and assuming $A^{(l)}(C \setminus Y, x^{(l-1)}) \neq 0$ for all $l$ (where $x^{(l-1)}$ is the input to the $l$-th block and $x^{(0)} = x$), there exists a unique sequence of minimal token-patches $\Delta_x W^{(l)}(Y)$ and exact bias-patches $\Delta_x b'^{(l)}(Y)$ that achieve strict functional equivalence across the entire stack.*

*That is, the output of the full network on the full context is functionally identical to the output of the patched network on the ablated context:*

$$T_{\mathbf{W}, \mathbf{b}'}(C, x) = T_{\mathbf{W} + \Delta_x \mathbf{W}(Y), \mathbf{b}' + \Delta_x \mathbf{b}'(Y)}(C \setminus Y, x). \tag{20}$$

*For each layer $l = 1, \ldots, L$, the exact bias-patch is given by the layer's context vector:*

$$\Delta_x b'^{(l)}(Y) := \delta A_{x^{(l-1)}}^{(l)}(Y), \tag{21}$$

*and the minimal token-patch for the weights, unique in that it strictly minimizes the Frobenius norm $\|\Delta_x W^{(l)}(Y)\|_F$ while satisfying pre-activation equivalence, is given by:*

$$\Delta_x W^{(l)}(Y) := \frac{(W^{(l)} \delta A_{x^{(l-1)}}^{(l)}(Y)) A^{(l)}(C \setminus Y, x^{(l-1)})^T}{\|A^{(l)}(C \setminus Y, x^{(l-1)})\|^2}, \tag{22}$$

*where $\delta A_{x^{(l-1)}}^{(l)}(Y) := A^{(l)}(C, x^{(l-1)}) - A^{(l)}(C \setminus Y, x^{(l-1)})$.*

*Furthermore, this unique minimal displacement update is conservative at every layer: $(W^{(l)} + \Delta_x W^{(l)}(Y))u = W^{(l)}u$ for all vectors $u$ orthogonal to the layer's query representation $A^{(l)}(C \setminus Y, x^{(l-1)})$.*

*Proof.* We first establish the result for a single layer $l$ given an arbitrary input $z$. We temporarily drop the layer index $l$ for notational clarity.

To achieve functional equivalence, we must match the pre-activation input to $g_\theta$. We seek a matrix $M$ such that $M A(C \setminus Y, z) = W \delta A_z(Y)$. Let $v = A(C \setminus Y, z)$ and $\delta = W \delta A_z(Y)$. We minimize the squared Frobenius

norm $\frac{1}{2}\|M\|_F^2$ subject to the linear constraint $Mv = \delta$. By introducing a vector of Lagrange multipliers $\lambda \in \mathbb{R}^d$, the Lagrangian is:

$$\mathcal{L}(M, \lambda) = \frac{1}{2}\operatorname{Tr}(M^T M) + \lambda^T(\delta - Mv). \tag{23}$$

Taking the gradient with respect to $M$ and setting it to zero yields $M = \lambda v^T$. Substituting this into the constraint $Mv = \delta$ gives $(\lambda v^T)v = \delta$, which implies $\lambda = \frac{\delta}{\|v\|^2}$. Thus, the unique minimal Frobenius norm token-patch is exactly:

$$\Delta_z W(Y) = \frac{\delta v^T}{\|v\|^2} = \frac{(W\delta A_z(Y))A(C \setminus Y, z)^T}{\|A(C \setminus Y, z)\|^2}. \tag{24}$$

Now, let us evaluate the full contextual block using the patched weights and patched biases on the ablated context $C \setminus Y$:

$$
\begin{aligned}
T_{W+\Delta_z W(Y),\, b'+\Delta_z b'(Y)}(C \setminus Y, z) \;=\;\; &\operatorname{LayerNorm}\Big(A(C \setminus Y, z) \\
&+ W' g_\theta\left((W + \Delta_z W(Y))A(C \setminus Y, z) + b\right) \\
&+ b' + \Delta_z b'(Y)\Big).
\end{aligned}
$$

By our construction of $\Delta_z W(Y)$, we know $\Delta_z W(Y)A(C \setminus Y, z) = W\delta A_z(Y)$. Substituting this into the pre-activation term:

$$
\begin{aligned}
(W + \Delta_z W(Y))A(C \setminus Y, z) + b \;=\;\; &W A(C \setminus Y, z) + W\delta A_z(Y) + b \\
=\;\; &W\left(A(C \setminus Y, z) + \delta A_z(Y)\right) + b \\
=\;\; &W A(C, z) + b.
\end{aligned}
$$

This perfectly matches the pre-activation of the unpatched network on the full context. Substituting this back, and replacing $\Delta_z b'(Y)$ with its definition $\delta A_z(Y)$, the block output becomes:

$$
\begin{aligned}
T_{W+\Delta_z W(Y),\, b'+\Delta_z b'(Y)}(C \setminus Y, z) \;=\;\; &\operatorname{LayerNorm}\Big(A(C \setminus Y, z) + W' g_\theta\left(W A(C, z) + b\right) + b' + \delta A_z(Y)\Big) \\
=\;\; &\operatorname{LayerNorm}\Big(\left(A(C \setminus Y, z) + \delta A_z(Y)\right) + W' g_\theta\left(W A(C, z) + b\right) + b'\Big).
\end{aligned}
$$

Because $A(C \setminus Y, z) + \delta A_z(Y) = A(C, z)$ by definition of the context vector, we finally get:

$$
\begin{aligned}
T_{W+\Delta_z W(Y),\, b'+\Delta_z b'(Y)}(C \setminus Y, z) \;=\;\; &\operatorname{LayerNorm}\Big(A(C, z) + W' g_\theta\left(W A(C, z) + b\right) + b'\Big) \\
=\;\; &T_{W,b'}(C, z).
\end{aligned}
$$

This confirms exact functional equivalence for a single block. (The conservative adaptation property $(W + \Delta_z W(Y))u = Wu$ for $u \perp v$ follows identically to the proof of Theorem 2.3.)

We now extend this to the stack of $L$ blocks via induction. Let $x^{(0)} = x$. Define the exact activations of the original network (with full context $C$) as $x^{(l)} = T^{(l)}(C, x^{(l-1)})$ for $l = 1, \ldots, L$. Similarly, define the activations of the modified network (with ablated context $C \setminus Y$ and updated parameters) as $\tilde{x}^{(l)} = T^{(l)}_{W^{(l)}+\Delta W^{(l)},\, b'^{(l)}+\Delta b'^{(l)}}(C \setminus Y, \tilde{x}^{(l-1)})$, with base input $\tilde{x}^{(0)} = x$.

For the base case $l = 1$, applying the single-layer result derived above with input $z = x^{(0)} = x$, we have:

$$\tilde{x}^{(1)} = T^{(1)}_{W^{(1)}+\Delta W^{(1)},\, b'^{(1)}+\Delta b'^{(1)}}(C \setminus Y, x^{(0)}) = T^{(1)}(C, x^{(0)}) = x^{(1)}.$$

For the inductive step, assume that the input to the $l$-th layer matches perfectly: $\tilde{x}^{(l-1)} = x^{(l-1)}$. We want to show that the output $\tilde{x}^{(l)}$ also matches $x^{(l)}$. By definition:

$$\tilde{x}^{(l)} = T^{(l)}_{W^{(l)}+\Delta W^{(l)},\, b'^{(l)}+\Delta b'^{(l)}}(C \setminus Y, \tilde{x}^{(l-1)}).$$

Using the induction hypothesis, we substitute $\tilde{x}^{(l-1)}$ with $x^{(l-1)}$:

$$\tilde{x}^{(l)} = T^{(l)}_{W^{(l)}+\Delta W^{(l)}, b'^{(l)}+\Delta b'^{(l)}}(C \setminus Y, x^{(l-1)}).$$

Applying the exact single-layer equivalence proven above for block $T^{(l)}$ with input $z = x^{(l-1)}$, we obtain:

$$T^{(l)}_{W^{(l)}+\Delta W^{(l)}, b'^{(l)}+\Delta b'^{(l)}}(C \setminus Y, x^{(l-1)}) = T^{(l)}(C, x^{(l-1)}) = x^{(l)}.$$

Thus, $\tilde{x}^{(l)} = x^{(l)}$ for all $l = 1, \ldots, L$. In particular, the final output of the stack is perfectly preserved: $\tilde{x}^{(L)} = x^{(L)}$, concluding the proof. $\square$

## C.2 An alternative implicit learning dynamics of ICL

In this section, we describe an alternate view on the implicit learning dynamics which follow from an iterative application of Theorem 2.3.

This approach differs in that it interprets how each context token input of a transformer affects the contextual block output. It's based on the idea that the influence of each context token on the model's output can be seen as an implicit change in its behavior. While the transformer's weights are not actually updated as it generates a response, the final output is effectively the same as if the model had undergone a rapid learning process influenced by the context. We will now describe this implicit learning dynamic.

Starting with the initial weight $W_0$ for the first dense layer of the neural network, we have

$$T_{W_0}(c_1, \ldots, c_n, x) = T_{W_0 + \Delta W_0(c_1)}(c_2, \ldots, c_n, x) \tag{25}$$

which gives us the first weight update corresponding on the effect of token $c_1$ on the first-layer weight matrix:

$$W_1 = W_0 + \frac{(W_0 \Delta A(c_1)) A(c_2, \ldots, c_n, x)^T}{\|A(c_2, \ldots, c_n, x)\|^2} \tag{26}$$

If we continue this process iteratively, we obtain the next weight update corresponding to the consumption of the second token:

$$T_{W_1}(c_2, \ldots, c_n, x) = T_{W_1 + \Delta W_1(c_2)}(c_3, \ldots, c_n, x) \tag{27}$$

which yields

$$W_2 = W_1 + \frac{(W_1 \Delta A(c_2)) A(c_3, \ldots, c_n, x)^T}{\|A(c_3, \ldots, c_n, x)\|^2} \tag{28}$$

We can summarize this iterative process of implicit weight updates for each successive token:

**Corollary C.3.** *In the notation above, the iterative process of weight updates*

$$W_i = W_{i-1} + \frac{(W_{i-1} \Delta A(c_i)) A(c_{i+1}, \ldots, c_n, x)^T}{\|A(c_{i+1}, \ldots, c_n, x)\|^2} \tag{29}$$

*starting with the initial weights of the first dense layer $W_0$ models the transfer of information from the prompt token $c_i$ into the contextual block weights: Namely, we have that*

$$T_{W_i}(c_{i+1}, \ldots, c_n, x) = T_{W_0}(c_1, \ldots, c_n, x), \tag{30}$$

*for all $i = 1, \ldots, n$ with $\Delta A(c_i) = A(c_i, \ldots, c_n, x) - A(c_{i+1}, \ldots, c_n, x)$.*

Notice that $\Delta A(c_i)$ measures the effect of context token $c_i$ on the contextual block output. When $c_i$ has no effect on the output, that is when $\Delta A(c_i)$ is zero, and the corresponding update vanishes. Notice that the weight update at step $i$ is linear in the weights; namely, we can rewrite it as

$$W_i = W_{i-1} + h_i W_{i-1} A_i = W_{i-1}(I + h_i A_i) \quad \text{where} \quad A_i := \Delta A(c_i) A(c_{i+1}, \ldots, c_n, x)^T \tag{31}$$

with adaptive learning rate given by

$$h_i := \frac{1}{\|A(c_{i+1}, \ldots, c_n, x)\|^2}. \tag{32}$$

In particular, this gives us a factorization formula for the total implicit weight matrix corresponding to the effect of context $[c_1, \ldots, c_n]$ on input-token $x$:

$$W_n = W_0(I + h_1 A_1)(I + h_2 A_2) \cdots (I + h_n A_n). \tag{33}$$

## D    Experiment Details

This section details the architectures, datasets, and training hyperparameters used to produce the results in Section 3.

### D.1    Task Definition: Linear Regression In-Context

Across all experiments, we generate instances of prompts composed of input-output pair token-vectors

$$[c_1, \ldots, c_n, z] \quad \text{with} \quad c_l = (x_k, h(x_k))^T \quad \text{and} \quad z = (x_{\text{query}}, 0)^T,$$

where $x_i, x_{\text{query}} \sim \mathcal{N}(0, I_d)$. The function $h(x) = \omega^T x$ is sampled such that $\omega \sim \mathcal{N}(0, I_d)$. We use feature dimension $d = 2$ and context length $N = 100$ (unless otherwise specified). The model predicts the scalar $\hat{y}(x_{\text{query}}) = \omega_{\text{test}}^T x_{\text{query}}$. We train using the MSE loss over batches of size $B = 32$.

### D.2    Multi-Layer Pre-LN Transformer Architecture

For the numerical verification in Section 3.1, we employ an autoregressive transformer matching the architecture described in Vaswani et al. (2017). The network consists of $L = 10$ blocks, where the $i$-th block is defined as:

$$T^{(i)}(x) = \text{LayerNorm}(y + \text{MLP}^{(i)}(y)), \quad \text{where} \quad y = \text{LayerNorm}(x + \text{MultiHeadAttn}^{(i)}(x)).$$

We set $d_{\text{mlp}} = 128$, $d_{\text{model}} = 32$, and use $h = 8$ attention heads with $d_k = d_{\text{model}}/h$. The model was trained using Adam optimizer with a learning rate of 0.001.

### D.3    Finetuning Comparison Setup

For the explicit finetuning experiments (Section 3.2), we utilize a single-layer baseline variant ($L = 1$). We initialize the transformer with the pretrained weights, then finetune using SGD, processing one example $\begin{pmatrix} x_i \\ \langle \omega_{\text{test}}, x_i \rangle \end{pmatrix}$ at a time. The weight matrix $W$ of the MLP layer is updated at each step. We use a tuned SGD learning rate optimized for the task to ensure a fair comparison against the implicit gradient step.

### D.4    RNN Architectural Setup

For the architectural diagnostics in Section 3.3, we construct a model $T = M_W \circ \text{RNN}$, replacing the multi-head self-attention block with a standard Recurrent Neural Network layer. To ensure a fair comparison, we match the parameter scale, setting the feature dimension of the RNN to $d_{\text{model}} = 64$ and $d_{\text{mlp}} = 128$. We trained the RNN-based model using a learning rate of 0.005 for $10,000$ steps.

