# OpenReview forum: "Learning without training: The implicit dynamics of in-context learnings"
_TMLR — Under review for TMLR_

### Review · Reviewer_rXVN · 2026-07-07

**Summary Of Contributions:**

This paper focuses on studying the implicit dynamics of in-context learning (ICL). The main result is Theorem 2.3, which says that the context injection to the input sequence is equivalent to adding a low-rank term to the weight. Section 2.2 further extends this to the case of a series of context sequences. The authors verify their theory through experiments.

**Audience:**

Yes

**Audience Explanation:**

There has been a huge interest in understanding the mechanism of in-context learning. This submission derives some formulas regarding the dynamics of ICL, and in particular formulates it as a gradient descent w.r.t. the loss given by Proposition 2.6, so could be of interest to audience in this area.

**Claims And Evidence:**

No

**Claims Explanation:**

The strength of this submission is that it gives the exact formula of replacing ICL with changes in the weight W, and the result can be applied to a sequence of context tokens. The weakness, however, is that the analysis stops there. The more important question not answered by this submission is, now that we have the formula, what insights can we get from this? How can the result help us improve ICL? For example, Proposition 2.6 shows that ICL can be formulated as stochastic gradient descents with a certain learning rate and loss, then can we tune this learning rate or loss to improve ICL?

I think this is a much more important question that needs to be answered, because there has already been a lot of work on deriving formulas for ICL, and what we need more is how we can improve ICL or perhaps more generally prompt tuning. For example, prior work (cited in this paper) has shown that ICL can be formulated as gradient descent, can learn the optimal weight in linear regression, etc. These papers have already derived formulas for ICL, so while the exact same result in this paper might not have appeared in prior work, the novelty is not really significant. These results all show the generality of ICL, that is it can do a lot of stuff, but they give us little insight into how to improve ICL.

Section 3.4 is a good start towards this direction. If the results can provide a way of doing prompt compression, then it will be much more useful. In its current form, the result seems similar to the large body of prior work, and does not provide a lot of extra insight.

Mathematically, the results in this submission do not need profound techniques. The high-level insight is the following: Given context tokens $c_1,c_2,\cdots$, we can find least-norm matrices $W_1,W_2,\cdots$ such that $W(x+c_1+c_2+\cdots) = (W+W_1+W_2+\cdots)x$. Replacing the sum here with self-attention gives the results in the submission. This result itself is not significant. The question that needs to be answered is what can we get from these matrices $W_1,W_2,\cdots$.

**Requested Changes:**

See the comments above. I think to be accepted, this paper needs more analysis on how the formulas derived can guide the improvement of ICL.

---

### Review · Reviewer_iqtS · 2026-07-12

**Summary Of Contributions:**

This paper studies the mechanism underlying in-context learning (ICL) by proposing an alternative interpretation of transformer computation. The central claim is that a transformer block consisting of a contextual layer (e.g., self-attention) followed by an MLP is mathematically equivalent to a context-free forward pass in which the MLP weights are modified by a minimal low-rank update determined by the context.

The contributions are as follows

**Interesting Theoretical Perspective** The paper presents a novel reformulation of contextual processing as an equivalent implicit modification of downstream weights rather than hidden representations. This viewpoint differs from existing work that focuses on explicit computation graphs or circuit analysis.

**Experimental verification of algebraic equivalence**  The numerical verification confirms that the patched model reproduces the original forward pass to numerical precision, which provides confidence that the derivations are implemented correctly.

**Audience:**

Yes

**Audience Explanation:**

The topic is timely and of broad interest to the machine learning community

**Claims And Evidence:**

No

**Claims Explanation:**

The submission provides clear and mathematically sound evidence for its core equivalence result: a context-dependent forward pass can be represented by a context-free pass with a minimal low-rank modification to the downstream affine weights. However, several broader claims are not yet supported as convincingly:

1. The experiments are largely restricted to synthetic linear-regression settings. This supports the theory in controlled models, but does not yet demonstrate that the same interpretation explains in-context learning in pretrained language models.

2. The attention-versus-RNN experiment is suggestive, but the claim that unstable implicit updates explain poorer RNN in-context learning is stronger than the evidence provided, since only a narrow architecture and task setting are evaluated.

**Requested Changes:**

**Expand the empirical evaluation** Demonstrate that the proposed interpretation generalizes beyond the toy setting.


**Clarify the significance of the gradient-descent formulation** Since Proposition 2.6 is an algebraic reformulation, explain more clearly why this specific gradient interpretation is meaningful beyond the fact that any additive update sequence admits such a representation.

**Improve discussion of the "Thought Patch".** Evaluate its robustness under more diverse prompts and out-of-distribution queries.

---

### Review · Reviewer_o99w · 2026-07-22

**Summary Of Contributions:**

This paper proposes a counterfactual reparameterization view of in-context learning. It introduces a "contextual block", consisting of a context-dependent layer followed by a neural network, and shows that, for a fixed context chunk $Y$ and query $x$, the effect of removing $Y$ can be exactly absorbed into the first affine layer of the subsequent network. The minimum-Frobenius-norm update is
$
\Delta_x W(Y)=\frac{
W\big(A(C,x)-A(C\setminus Y,x)\big)
A(C\setminus Y,x)^\top
}{
|A(C\setminus Y,x)|_2^2
},
$
which has rank at most one. The paper extends this construction to stacked blocks with residual connections, rewrites the sequence of context-dependent patches as a gradient-update-like dynamics, and gives a specially constructed linear-attention example in which the resulting expression coincides with one step of least-squares gradient descent. Experiments on synthetic in-context linear regression numerically verify the algebraic identity, compare the constructed patches with explicit finetuning, contrast attention and RNN patches, and explore a query-independent least-squares approximation called a "Thought Patch"

A strength of the paper is that the minimum-norm reparameterization is stated clearly and the distinction between an actually executed update and a counterfactual equivalent computation is acknowledged. However, my main concern is that the central theorem is a generic and largely immediate linear-algebraic construction, whereas the paper repeatedly interprets it as an explanatory learning mechanism for ICL. The theory does not establish that the constructed updates correspond to a natural task objective, are produced through model training, follow a common descent direction, or possess convergence or generalization properties. In my view, this gap between the proved statement and the broader interpretation is fundamental.

**Additional Comments:**

None

**Audience:**

Yes

**Audience Explanation:**

The mechanisms underlying ICL, the relationship between activation changes and parameter-space updates, and context compression are all relevant topics for a portion of the TMLR audience. Even when interpreted narrowly, the minimum-norm counterfactual patch may be a useful algebraic or diagnostic perspective, and the connection to activation task vectors and low-rank editing is potentially interesting.

However, audience interest alone does not resolve the first TMLR criterion. In its current form, the paper presents the construction as a general explanation of implicit learning and ICL, whereas the evidence primarily supports a query-conditioned functional reparameterization.

**Broader Impact Concerns:**

I do not identify a broader-impact or ethical concern that would require an additional statement.

**Claims And Evidence:**

No

**Claims Explanation:**

The exact functional-equivalence statement is mathematically valid, but I do not find the paper’s central claims about "implicit learning dynamics" and the mechanism of ICL adequately supported.

First, Theorem 2.3 reduces to the standard minimum-norm problem. This construction applies to essentially any context-dependent representation map followed by an affine layer. The rank-one structure follows because the update is required to match the desired change at only one input vector. It therefore does not, by itself, identify a transformer-specific mechanism, a learned algorithm, or a structural property that distinguishes successful ICL models from arbitrary contextual systems.

Second, the exact patch is both query-dependent and counterfactual. Computing $\Delta_xW(Y)$ requires access to both $A(C,x)$ and $A(C\setminus Y,x)$, and changing $x$ generally changes the patch. Thus, the theorem does not show that a context is converted into a single reusable weight update. It shows that, after fixing one context and one query, an analyst can construct a matrix reproducing that particular activation difference. This considerably weakens statements that the patch “represents the context” or explains adaptation to new inputs. The static Thought Patch in Section 3.4 is a separate least-squares approximation, for which no approximation or out-of-distribution generalization guarantee is established.

Third, Proposition 2.6 does not establish a substantive gradient-descent mechanism. As Remark 2.7 itself acknowledges, any additive sequence $W_{i+1}=W_i+G_i$ can be written as gradient descent on an artificially defined linear loss. Here, the losses $L_i$ are constructed retrospectively from the desired updates, vary across tokens, and need not correspond to a common task objective. The paper does not prove that:

* the gradient is that of a natural downstream or pretraining objective;
* the same objective is decreased across steps;
* the dynamics arises from the model’s training process;
* the updates converge to a meaningful solution; or
* the resulting predictor generalizes.

The least-squares example does not resolve this issue. It manually selects the linear-attention and output weights so that the algebra reproduces one full-batch gradient step at the initial parameter. This is an expressivity or existence construction, rather than a result showing that gradient-based pretraining learns these weights or that the proposed dynamics emerges under stated training assumptions.

Fourth, the empirical evidence is insufficient for the broader mechanistic claims. Figure 1 primarily verifies an identity that holds by construction. Figure 2 studies a single synthetic linear-regression setting and shows alignment with a tuned explicit finetuning procedure, but does not establish that the same phenomenon holds across tasks, model classes, initializations, or pretrained language models. Figure 3 does not justify the causal conclusion that RNNs have weaker ICL because their constructed patch increments do not converge: no theorem or controlled experiment connects this diagnostic quantity to prediction performance. Similarly, the Thought Patch experiments do not evaluate actual prompt-compression quality, memory reduction, latency, autoregressive error accumulation, or realistic language tasks.

Finally, the related-work positioning is incomplete. The paper discusses several foundational implicit-gradient and task-vector works, but does not sufficiently compare its counterfactual, query-specific construction against more recent analyses that address task-concept retrieval, semantic task representations, actual training dynamics, the competition between in-context and in-weight learning, latent-variable inference, and the limitations of reusable task vectors. These distinctions are particularly important because those works study several of the substantive questions left open here: why training produces an ICL algorithm, what task-level object is retrieved, and under what conditions it generalizes.

My negative assessment is therefore not based merely on the modest size of the mathematical result. Rather, the main explanatory claims go substantially beyond what the proved reparameterization identity and the current experiments establish.

**Requested Changes:**

### Critical changes

1. The paper should consistently distinguish a post-hoc functional reparameterization from a mechanism that the transformer learns or executes. Claims that contextual blocks “perform implicit finetuning” that the construction “uncovers an implicit gradient descent learning dynamics” or that it explains why LLMs perform ICL should either be significantly weakened or supported by additional theory.

2. A substantive theoretical advance would require explicit data, architecture, and training assumptions under which the constructed update approximates the gradient of a fixed, natural task loss. Ideally, the analysis should also establish a descent property, convergence, task recovery, prediction error, or generalization guarantee. Merely defining a different linear loss for each arbitrary additive update is not sufficient.

3. **Explain how the proposed structure is learned.**
   The current least-squares example hand-designs weights that implement one gradient step. The paper should analyze whether and how gradient-based pretraining converges to such weights, or clearly present the result only as an expressivity construction. This issue is central to the paper’s stated goal of explaining emergent ICL.

4. The exact patch $\Delta_xW(Y)$ is not a context-only update. The paper should either derive conditions under which one context-dependent patch works uniformly over a nontrivial query class, with approximation and generalization bounds, or explicitly state throughout that the exact theorem is pointwise in $x$. Claims about identical behavior on “new inputs” should be corrected unless the patch is recomputed for each new input.

5. Experiments should test pretrained transformers on multiple nontrivial ICL tasks and should include interventions or predictions that would differ under competing explanations. Relevant evaluations include patch/finetuning-gradient alignment across layers, models, tasks and seeds; comparisons against random or matched-rank patches; causal interventions; and tests of whether the patch predicts ICL success or failure.

6. The paper should compare the exact questions answered by its pointwise reparameterization with work on learned ICL algorithms, training dynamics, task-vector emergence and limitations, semantic/task-concept retrieval, latent-variable interpretations, and in-context versus in-weight learning. Representative relevant works include those listed below. I do not regard citation of every item as mandatory, but the conceptual distinctions they represent need to be addressed.

### Changes that would strengthen the paper

- Clarify that the uniqueness in Theorem C.2 is uniqueness within the chosen layerwise pre-activation-matching and minimum-norm construction, rather than uniqueness among all possible cross-layer parameter modifications producing the same final output.

- Clarify the architecture terminology. The architecture called “Pre-LN” in Appendix D appears to apply LayerNorm after the residual addition; this differs from the usual Pre-LN convention.

- Report results over independent training runs and provide uncertainty for Figures 2–4, release code, and state how finetuning hyperparameters and calibration queries were selected.

- Correct minor presentation issues, including the title’s "in-context learnings", incomplete figure captions, and several grammatical errors.

### Representative related work (worth explicit discussion)

* Bu et al. (NeurIPS 2024). *Provably Transformers Harness Multi-Concept Word Semantics for Efficient In-Context Learning.*

* Chan et al. (ICLR 2025). *Toward Understanding In-Context vs. In-Weight Learning.*

* Mittal et al. (ICML 2025). *Does Learning the Right Latent Variables Necessarily Improve In-Context Learning?*

* Bu et al. (ICML 2025). *Provable In-Context Vector Arithmetic via Retrieving Task Concepts.*

* Panwar et al. (ICLR 2024). *In-Context Learning through the Bayesian Prism.*

* Dong et al. (ICLR 2026). *Understanding Task Vectors in In-Context Learning: Emergence, Functionality, and Limitations.*